# Application of the Standardised Streamflow Index for Hydrological Drought Monitoring in the Western Cape Province, South Africa: A Case Study in the Berg River Catchment

Mxolisi Blessing Mukhawana [1,2,*], Thokozani Kanyerere [2], David Kahler [3] and Ndumiso Siphosezwe Masilela [1]

1   Department of Water and Sanitation, South African Government, Pretoria 0001, South Africa; masilelan2@dws.gov.za
2   Department of Earth Sciences, University of the Western Cape, Cape Town 7535, South Africa; tkanyerere@uwc.ac.za
3   Center for Environmental Research and Education, Duquesne University, Pittsburgh, PA 15282, USA; kahlerd@duq.edu
*   Correspondence: mukhawanam@dws.gov.za

**Abstract:** In many regions around the world, drought has been recurrent, more frequent, and more intense over time. Hence, scientific research on drought monitoring has become more urgent in recent years. The aim of this study was to test the applicability of the Standardised Streamflow Index (SSI) for hydrological drought monitoring in the Berg River catchment (BRC), Western Cape (WC) province, South Africa (SA). Using various methods described in this study, the sensitivity of the SSI to the commonly used Gamma, Log-normal, Log-logistic, Pearson Type III, and Weibull Probability Distribution Functions (PDFs) was tested. This study has found that all the tested PDFs produced comparable results for mild to severe drought conditions. The SSI calculated using the Gamma, Log-Normal, and Weibull PDFs is recommended for the BRC because it consistently identified extreme drought conditions during the 1990–2022 study period and identified the 2015–2018 droughts as the worst during the study period. Although more studies are required to test other PDFs not considered, this study has shown that the SSI can be applicable in the BRC. This study has provided a foundation for more research on the application of the SSI in the BRC and other catchments in SA.

**Keywords:** hydrological drought; Standardised Streamflow Index (SSI); Standardised Precipitation Index (SPI)



## 1. Introduction

Historical records indicate that in most climate zones or regions around the world, drought has been recurrent, more frequent, and more intense. There is evidence of negative environmental and socio-economic impacts because of the recurring drought events [1]. Since 1976, the United Kingdom (UK) has experienced several severe droughts that resulted in serious water shortages on a national scale, negatively affecting mainly the agriculture and commerce industries [2]. It was reported that the United States of America (USA) lost over 10 billion dollars in damages because of the droughts that occurred during the year 2002 alone [3]. In South Africa (SA), the drought that occurred in 1992 was judged to be the most severe since the beginning of the 20th century. The resulting water shortages were responsible for crop and livestock losses in the agriculture and farming industries, as well as food shortages for people. During the year 2015, the droughts that occurred in the Western Cape (WC) province in SA were reportedly the most severe in just over a century. In response to these severe droughts, increased water use restrictions were enforced by the WC government [4–7]. Projections on climate variability (change) indicate that most regions

worldwide, including SA, will continue to experience more frequent and more intense drought events [8–12]. Hence, scientific research on the development of improved drought monitoring and early warning systems has become more urgent in recent years. If well developed and applied effectively, these systems have the potential to reduce vulnerability to drought impacts and may contribute to the development and implementation of suitable policies for improved drought management in SA [1,2].

The complex nature of drought has resulted in a lack of consensus on its definition. However, it is accepted worldwide that drought occurs in four phases: meteorological drought, agricultural drought, hydrological drought, and socio-economic drought. Both of these phases of drought are generally caused by prolonged deficits in rainfall, affecting soil moisture (meteorological), crop growth (agricultural), surface and ground water storages (hydrological), and the availability of water for human consumption (socio-economic) [5,6,12–18]. The description of drought according to its propagation phases has led to the development of important indices for monitoring drought around the world. Many indices have been developed and used for many years to monitor droughts and develop drought monitoring and early warning systems. They have commonly been used to characterize droughts according to their onset, duration, magnitude, frequency, end, and spatial coverage. The Standardised Precipitation Index (SPI) is one of the most widely used drought indices around the world. It is recommended by the World Meteorological Organization (WMO) as the preferred index for meteorological drought monitoring [4,19–23]. In the UK, the SPI has been used to characterize meteorological droughts to improve understanding of the nature of droughts and their related hazards [20,21]. In SA, the SPI has been thoroughly tested to characterize drought according to its duration, frequency, severity, intensity, and spatial extent in all the provinces and climatic regions [1,5,24–31]. The SPI is widely used across the world and especially in SA because it uses a simple calculation procedure, requires only rainfall data to calculate, and is flexible in that it allows the use of various time scales for monitoring different types of droughts [1,20,32–34]. However, the SPI has an inherent limitation that cannot be overlooked. Its use of only precipitation in its calculation procedure means that it is not capable of providing hydrological drought information that describes the direct impact of drought on surface and groundwater storages [1]. Hence, other indices should be considered for hydrological drought monitoring in SA.

To characterize hydrological drought according to its onset, duration, magnitude, frequency, and spatial coverage, the WMO has recommended the Standardised Streamflow Index (SSI), which uses the same calculation procedure as the SPI [1,35]. The SSI inherits the advantages of the SPI in that it uses a simple calculation procedure. It differs from the SPI in that it uses streamflow instead of rainfall data in its calculation procedure. Although the SSI does not incorporate the impact of water use or demand, its use has increased since its introduction. It has been tested and has performed well in various regions with different catchment characteristics around the world. It has proved useful for the characterization of hydrological droughts and the development of early warning systems in Slovenia, China, the UK, Azerbaijan, and Iran [2,19,36–39]. In SA, the SSI has been used to characterize hydrological drought in all the cape provinces [38]. The calculation simplicity of the SPI is enhanced by the fact that it has been extensively used around the world to the extent that the Gamma Probability Distribution Function (PDF) has been widely accepted for its calculation. On the other hand, the SSI has not been tested thoroughly enough to be able to reach a consensus on the universal PDF for its calculation. This consensus may not easily be reached because many or all catchments possess high spatial streamflow variability, resulting in high levels of uncertainty in the PDFs that fit streamflow data best [1,20,40]. The Gamma PDF was used to calculate the SSI and tested at varying catchments in SA, Slovenia, China, the Netherlands, and Iran [19,37,38,41,42]. The Generalized Extreme Value (GEV), Log-Logistic, and Tweedie PDFs were recommended to calculate the SSI and were tested in various catchments in the UK [2,20,40]. The GEV and the Log-Logistic PDFs were recommended for calculating the SSI and were tested in catchments in Spain [40]. The Tweedie, GEV, and Log-Logistic PDFs were recommended for calculating the SSI and were

tested in various catchments in Europe [40,43]. The above findings show that the best-fitting PDFs may vary with varying catchments. This is supported by Li et al. (2018) who tested the Pearson Type III (PTIII), Log-Logistic, GEV, and Log-Normal and concluded that the best fitting PDFs varied with varying catchments and streamflow gauging station locations [44]. Hence, according to the above studies, the Gamma PDF, as used by Botai et al. (2021) and others, may not be the most suitable candidate for calculating the SSI in some SA catchments [20,40,43,44]. Hence, other PDFs should be tested in SA catchments.

Although it is evident that drought affects surface water supply systems such as rivers, there are very few research studies on the use of the SSI to characterize droughts in SA [38]. Consequently, there is no consensus on the accepted PDFs for SSI calculation in the various catchments in SA. Hence, the aim of this study is twofold: to evaluate the applicability of the SSI for hydrological drought monitoring in SA and to test the sensitivity of the SSI to different PDFs at a selected catchment in the WC province of SA, potentially leading to recommendations or guidelines for the selection of the most suitable or best-fitting PDFs in other SA catchments with varying geo-hydro-climatic zones.

The WMO has recommended the SSI for hydrological drought monitoring, but the SSI needs to be thoroughly tested at various catchments in SA. Therefore, the results from this study will contribute to the provision of tested scientific knowledge on the effective application of the SSI for hydrological drought monitoring in SA. Given the few studies that have been conducted on the application of the SSI in SA, the overall outcomes from this study will provide a foundation and a basis for the application of the SSI in SA. This may ultimately aid in the improvement of drought monitoring and early warning systems in SA. The remainder of the paper is organized as follows: Section 2 presents the case study area and the data used in the study. Section 3 briefly describes the methods and approaches used. Section 4 presents key results and the discussion. Section 5 presents the main conclusions of this study.

## 2. Study Area and Data

### 2.1. Study Area

This study was carried out on the approximately 7700 km$^2$ Berg River Catchment (BRC) (Figure 1), one of the two catchments in the Berg-Olifants Water Management Area (WMA). The BRC supplies water to parts of the WC province in SA. As shown in Figure 1, the Berg River in the BRC forms at the Franschhoek mountains and flows northwards, where it is joined by the Klein Berg, and then flows westwards until it discharges into the Atlantic Ocean. With a total length of approximately 285 km, the Berg River has up to nine major and seven minor tributaries. Six of these minor tributaries, which include the Klein Berg River, are perennial [1,5]. Surface water is a major water source in the BRC. The Mean Annual Runoff (MAR) in the upper Berg River and its tributaries is approximately $277 \times 106$ m$^3$, approximately $263 \times 106$ m$^3$ at the upper middle Berg River and its tributaries, approximately $288 \times 106$ m$^3$ at the lower middle Berg River, $97 \times 106$ m$^3$ at the lower Berg River and its tributaries, and approximately $17 \times 106$ m$^3$ at the flood plain and estuary [1,5,45]. Thus, monitoring hydrological drought using streamflow is crucial in the BRC. The WC province experiences both winter, summer, and all-year rainfall. The annual rainfall in the WC ranges between 300 mm and 900 mm. The BRC is situated in the winter rainfall zone of the WC province, with a maximum rainfall of approximately 30 mm in June [1,5,45].

### 2.2. Streamflow Data

The monthly mean streamflow data from 1990 to 2022 used in this study to calculate the SSI were obtained from the South African National Department of Water and Sanitation (DWS) (Figure 1). The streamflow data considered were assumed to be from near-natural flow rivers. In this study, the authors used data from three streamflow gauging stations that met the minimum required record of 30-years for calculating the SSI and had minimal missing data or gaps. The G1H020 is in the upper Berg River, with a MAR of approximately $277 \times 106$ m$^3$. The G1H013 is in the upper middle Berg River, with a MAR of approximately

$263 \times 10^6$ m$^3$. The G1H008 is in the Klein Berg River on the lower middle Berg River, with a relatively low MAR of $263 \times 10^6$ m$^3$. The Klein Berg River is a tributary of the Berg River (Figure 1 and Table 1). An assessment of the historical data obtained from G1H020, G1H013, and G1H008 indicates that those located on the middle and upper Berg River record relatively higher flows than those located on the Klein Berg River (Figure 2). Hence, in this study, to test the sensitivity of the SSI to various PDFs, streamflow time series were acquired from three streamflow gauging sites: G1H008, located on the low flow Klein Berg River; G1H013 located on the relatively high-flow lower part of the Berg River; and G1H020 also located on the relatively high-flow upper part of the Berg River.

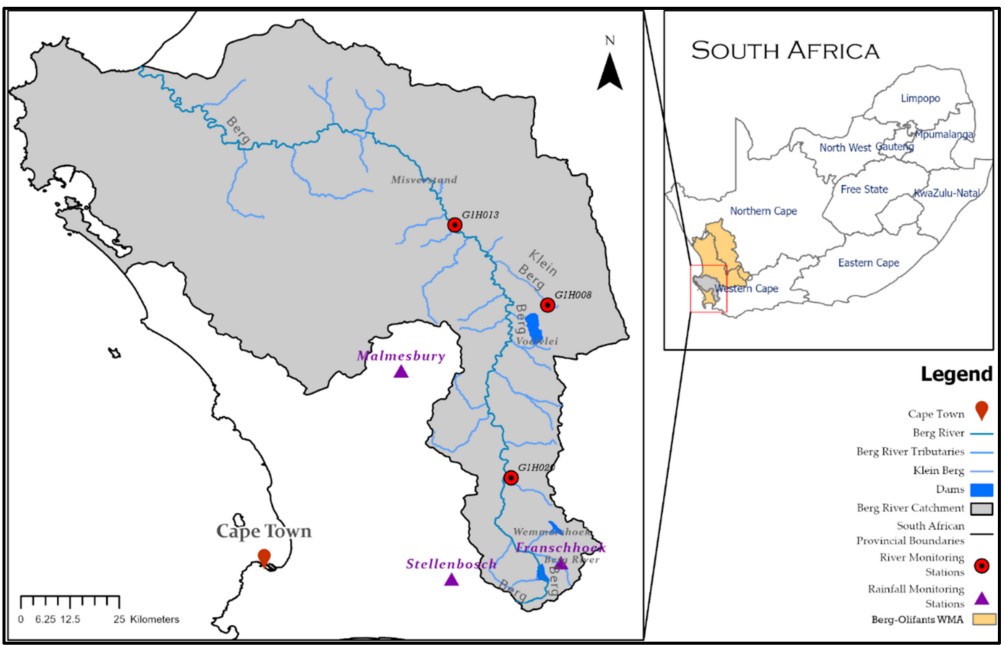

**Figure 1.** The Berg River Catchment and the selected Streamflow and rainfall monitoring stations.

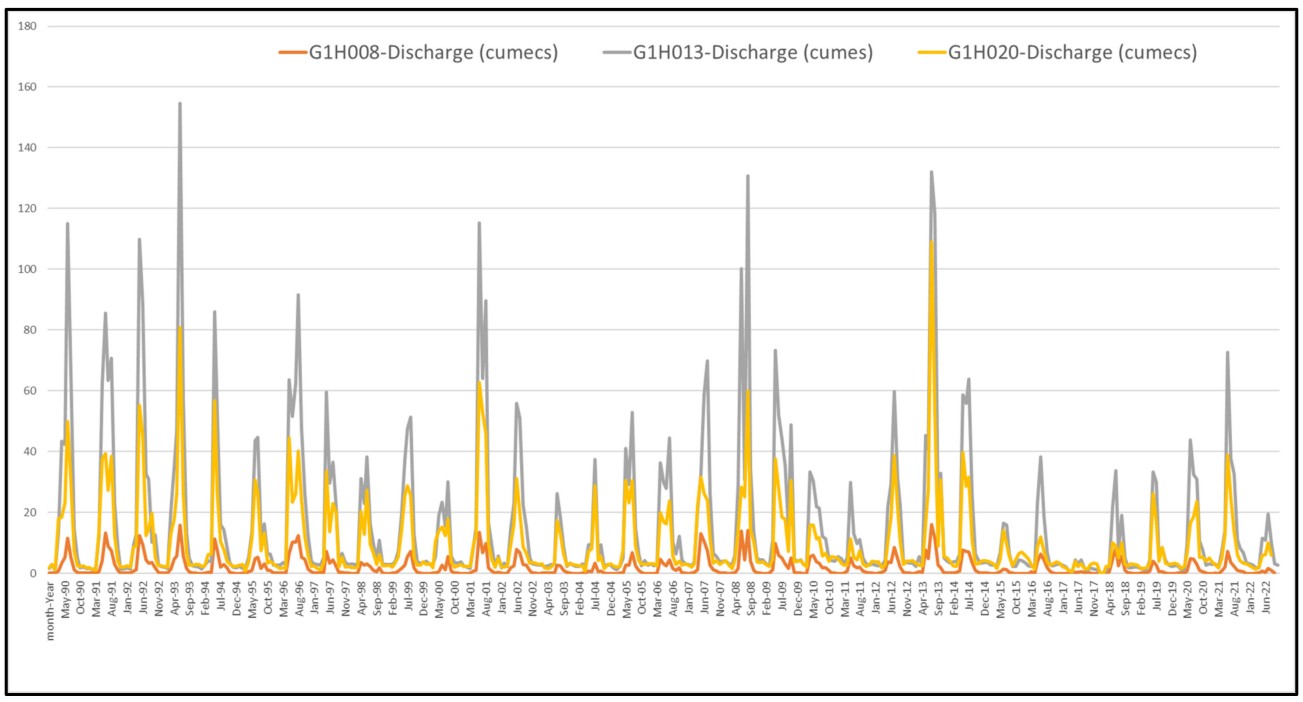

**Figure 2.** Historical streamflow patterns (1990–2022) at gauging stations G1H008, G1H013, and G0H020 on the Berg and Klein Berg rivers in the Berg River Catchment (m$^3$/s $\cong$ cumecs).

**Table 1.** Streamflow gauging stations that were used to obtain river discharge data for SSI calculations in the Berg River Catchment.

| Streamflow Gauging Station Identity | River | Location Coordinates (Latitude: Longitude) | Period (Years) |
|---|---|---|---|
| G1H008 | Klein Berg | −33.313889:19.074722 | 1990 to 2022 (32 Years) |
| G1H013 | Berg | −33.130833:18.862778 | 1990 to 2022 (32 Years) |
| G1H020 | Berg | −33.707778:18.991111 | 1990 to 2022 (32 Years) |

*2.3. Rainfall Data*

The monthly mean rainfall data from 1980 to 2021 were used in this study to calculate the SPI was obtained from the South African Weather Service (SAWS). Although all three stations contained rainfall data that met the requirement of a minimum of a 30-year record, only the Franschoek and Stellenbosch stations were used in this study because they had more recent data (Figure 3). Hence, in this study, the rainfall data used were obtained from two gauging stations located in the Franschoek and Stellenbosch towns (Figure 1) because the data met the minimum required record of 30 years for calculating the SPI.

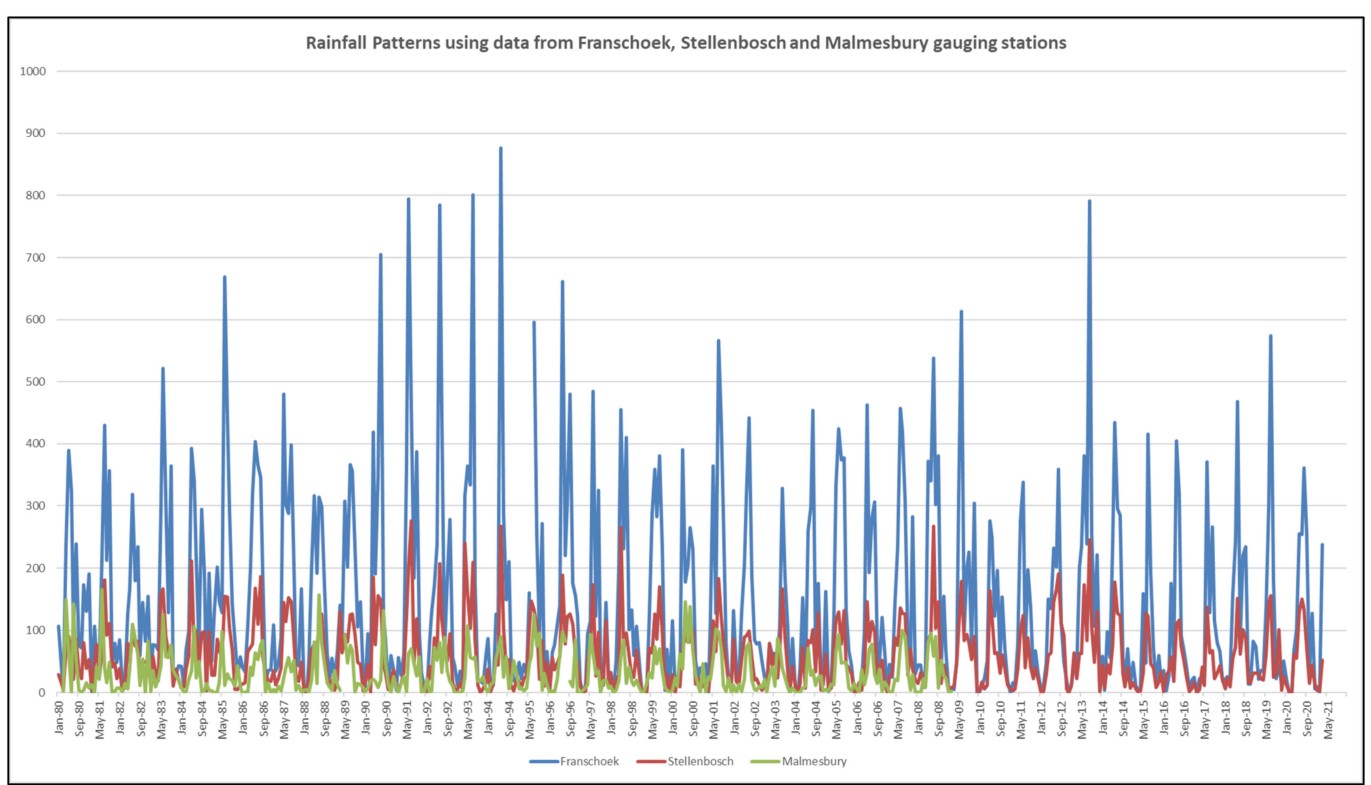

**Figure 3.** Historical rainfall (mm) (1980–2021) patterns in Franschoek, Stellenbosch, and Malmesbury towns, located in and around the Berg River Catchment.

## 3. Methods

*3.1. SSI Calculation*

In this study, the SSI was computed using streamflow data obtained from the G1H020, G1H013, and G1H008 streamflow gauging stations located in the BRC, as well as various commonly used PDFs. The resultant SSI obtained from the PDF that best fitted the streamflow data was used to characterize hydrological drought in the BRC. The simple SPI-based generic procedure was followed for calculating the SSI: (1) monthly streamflow time series were averaged over 12 months' time scales; (2) a PDF was fitted to the streamflow time series using 12 months' time scales; (3) the PDF's parameters were determined using the

streamflow data; (4) cumulative distribution functions were established for the streamflow and used to calculate the cumulative probability of the observed values of the variables; and (5) the inverse normal cumulative distribution function with a mean of zero and variance of one was applied to generate the SSI12 time series. In the SSI12 series, the zero values are equivalent to the mean streamflow. Negative values indicate dryer than average conditions, while positive values indicate wetter than average conditions [1,19,20,32–34,40]. The SSI is therefore calculated using Equation (1) [40]:

$$SSI = W - \frac{C_0 + C_1 W + C_2 W^2}{1 + d_1 W + d_2 W^2 + d_3 W^3} \tag{1}$$

where $W = \sqrt{-2\ln(P)}$ for $p \leq 0.5$.

$p$ is the probability of exceeding a determined x value, and $p = 1 - F(x)$. If $p > 0.5$, $p$ is replaced by $1 - p$, and the sign of the resultant $SSI$ is reversed. $C_0$ = 2.515517; $C_1$ = 0.802853; $C_2$ = 0.010328; $d_1$ = 1.432788; $d_2$ = 0.189269; and $d_3$ = 0.001308 are constants. If the PDF, F(x), is suitable for fitting the monthly streamflow series, the average value of the SSI and the standard deviation must equal 0 and 1, respectively [40].

Drought classification using the SSI may differ for various studies. The drought classification used in this study is described in Table 2 [1,1,19,20,32–34,40].

**Table 2.** SSI and SPI drought Classification.

| SPI/SSI Values | Drought Classification |
| :---: | :---: |
| $\geq$2.00 | Extremely Wet |
| 1.50 to 1.99 | Severely Wet |
| 1.00 to 1,49 | Moderately Wet |
| 0.00 to 0.99 | Mildly Wet |
| 0.00 to $-$0.99 | Mild Drought |
| $-$1.00 to $-$1.49 | Moderate Drought |
| $-$1.5 to $-$1.99 | Severe Drought |
| $\leq$$-$2.00 | Extreme Drought |

*3.2. PDFs Considered for SSI Calculation*

To evaluate the applicability of the SSI for hydrological drought monitoring in the BRC, the sensitivity of the SSI to various commonly used PDFs was tested. Five PDFs, i.e., Gamma, Log-normal, Pearson Type III (PTIII), Log-Logistic, and Weibull (Equations (2)–(6)), were fitted to the streamflow time series obtained from G1H008, G1H013, and G1H020 in the BRC (Table 3). The Gamma PDF was selected because it is the commonly preferred method for calculating the SPI and its performance for calculating the SSI has not been thoroughly tested in SA catchments. The only record that could be found on the SSI application in SA using the Gamma PDF is by Botai et al. (2021) [38]. The Log-normal, PTIII, Log-Logistic, and Weibull PDFs were selected because they have been tested in European and other catchments but not in SA catchments. No records were found on the SSI application in SA using the Log-normal, PTIII, Log-Logistic, and Weibull PDFs. It is recommended that other PDFs be tested in future research. The Log-normal and Gamma are two-parameter PDFs, while the PTIII, Log-Logistic, and Weibull are three-parameter PDFs. Only the Lognormal and Gamma are bound below zero. Following the approaches by Tijdeman et al. (2020) and Stagge et al. (2015), the L moments (Lmom) were used to estimate parameters of the PDFs [20,40,43,45]. Alternative parameter estimation methods may be considered for future studies. The selected PDFs and L moments have been commonly used and thoroughly tested for SPI and SSI calculation in many regions around the world, but not in RSA, especially for SSI calculation. The use of SSI is relatively new in RSA, so the focus of this study was to introduce the SSI and test it using currently commonly used

and relatively easy-to-apply PDFs and parameter selection methods. Follow up studies should consider other PDFs not tested in this study as well as other parameter selection methods. The Log-logistic, Log-Normal, PTIII, Weibull, and Gamma PDFs are determined using Equations (2)–(6), as shown in Table 3.

**Table 3.** Probability Distribution Functions used to calculate the SSI in the BRC [34,38,40,46].

| Probability Distribution Function (PDF) Used for SSI Calculation in the BRC | PDF Equations | |
|---|---|---|
| Log-logistic [40,46] | $F(x) = \left[ 1 + \left( \frac{\alpha}{x-\gamma} \right)^\beta \right]^{-1}$ | (2) |
| | $\beta = \frac{2w_1 - w_0}{6w_1 - w_0 - 6w_2}, \alpha = \frac{(w-2w_1)\beta}{\Gamma\left(1+\frac{1}{\beta}\right)\Gamma\left(1-\frac{1}{\beta}\right)}, \gamma = w_0 - \alpha\Gamma\left(1+\frac{1}{\beta}\right)\Gamma\left(1-\frac{1}{\beta}\right)$ | |
| Log-Normal [40,46] | $F(x) = \theta\left( \frac{ln\ (x-a)-\mu}{\sigma} \right)$ | (3) |
| | $\theta \approx$ standard normal cumulative distribution function. $\sigma = 0.999281z - 0.006118z^2 + 0.000127z^5$ such that $z = \sqrt{\frac{8}{3}\theta^{-1}}\left(\frac{1+\tau_3}{2}\right)$. $\mu = ln\left[\frac{\varepsilon_2}{\text{erf}\left(\frac{\sigma}{2}\right)}\right] - \frac{\sigma^2}{2} erf$ is the Gauss error function such that $\text{erf}\left(\frac{\sigma}{2}\right) = 2\theta\left(\frac{\sigma}{2}\sqrt{2}\right) - 1$ and $a = \varepsilon_1 - e^{\mu + \frac{\sigma^2}{2}}$. | |
| Pearson Type III [40,46] | $F(x) = \frac{1}{\alpha\Gamma(\beta)} \int_\gamma^x \left( \frac{x-\gamma}{\alpha} \right)^{\beta-1} e^{-\left(\frac{x-\gamma}{\alpha}\right)}$ | (4) |
| | If $\tau_3 \geq \frac{1}{3}$, then $\tau_m = 1 - \tau_3$, leading to $\beta = \frac{\left(0.36067\tau_m - 0.5967\tau_m^2 + 0.25361\tau_m^3\right)}{\left(1 - 2.78861\tau_m + 2.56096\tau_m^2 - 0.77045\tau_m^3\right)}$ If $\tau_3 < \frac{1}{3}$, then $\tau_m = 3\pi\tau_3^2$; such that $\beta = \frac{(1+0.2906\tau_m)}{\left(\tau_m + 0.1882\tau_m^2 + 0.0442\tau_m^3\right)}, \alpha = \sqrt{\pi}\varepsilon_2 \frac{\Gamma(\beta)}{\Gamma\left(\beta+\frac{1}{2}\right)}$ and $\gamma = \varepsilon_1 - \alpha$ | |
| Weibull [40,46] | $F(x) = 1 - e^{-\left(\frac{x-m}{a}\right)^b}$ | (5) |
| | $b = \frac{1}{(7.859C + 2.9554C^2)}, C = \frac{2}{3-\tau_3} - 0.6309, a = \frac{\varepsilon_2}{G\left(1+\frac{1}{b}\right)\left(1-2^{-\frac{1}{b}}\right)} m = \varepsilon_1 - a\Gamma\left(1+\frac{1}{b}\right)$ | |
| Gamma [34,38] | $g(x, \alpha, \beta) = \frac{1}{\beta^\alpha\ \Gamma(\alpha)}\left( x^{\alpha-1} e^{-\frac{x}{\beta}} \right)$ | (6) |
| | $\alpha > 0$ and $\beta > 0$ are the estimated shape and scale parameters, $x > 0$ is the streamflow ($m^3/s$), and $\Gamma(\alpha)$ is the Gamma PDF such that, $\Gamma(\alpha) = \int_0^\infty x^{\alpha-1} e^{-x} dx$. | |

### 3.3. SSI Computation Using R Software Package

The SSI calculations for G1H008, G1H013, and G1H020 streamflow time series were carried out using the R software package. The SSI time series computation using the Gamma, Log-logistic, and PTIII PDFs was carried out in R-Studio software using the SPEI version 1.7 package. The manual for the SPEI version 1.7 package was obtained from https://cran.r-project.org/web/packages/SPEI/index.html (accessed on 9 January 2023). The SSI time series computation using Weibull and Log-Normal PDFs was carried out in R-Studio software using the SCI version 1.0–2 package. The manual for the SCI version 1.0–2 package was obtained from https://www.rdocumentation.org/packages/SCI/versions/1.0-2 (accessed on 9 January 2023).

### 3.4. Evaluation of Best Fitting PDFs for SSI Computation

According to Svensson et al. (2017), the S-W test has been found to be the most powerful test for normality, closely followed by the Anderson-Darling and the Kolmogorov-Smirnov tests [20]. Thus, the Shapiro-Wilk (S-W) goodness-of-fit or normality test was used in this study to evaluate the sensitivity of the SSI to the selected PDFs. As used in the study by Svensson et al. (2017), the significance level chosen for this study is 95% (*p*-value = 0.05). The S-W test was applied to test the null hypothesis (H0) that the SSI time series is normally distributed. Thus, if the *p* value is less than or equal to 0.05, the null hypothesis is rejected,

and the time series is not normally distributed. If the *p* value is greater than 0.05, the null hypothesis is not rejected, and the time series used is normally distributed [20]. The S-W test helps to assess how well the considered PDFs fit the streamflow time series, resulting in an SSI time series that closely resemble the expected standard normal distribution.

### 3.5. Evaluation of the Correlation between the SSI Computed Using the Selected PDFs

Correlation coefficients are descriptive statistics used to describe the magnitude and direction of the relationship between variables. The correlation coefficients vary from $-1$ to $+1$, whereby the sign describes the direction of the relationship (positive or negative). When the coefficient is 0 or close to 0, there is little to no relationship. However, the closer the coefficient of correlation is to $-1$ or $+1$, the stronger the relationship is between the variables [47]. In this study, the Pearson's correlation coefficient described by Sedgwick (2012) is used to determine the linear relationship between the different probability distribution functions [48].

### 4. Results

#### 4.1. SSI Calculation Using the Selected PDFs

The results of the SSI calculation using the Gamma, Log-Logistic, Log-Normal, PTIII, and Weibull PDFs are shown in Figures 4–6. In Figure 4 (G1H008), it can be observed that the SSI12 time series computed using the Gamma, Log-Logistic, Log-Normal, PTIII, and Weibull PDFs produced drought events with similar onset and end times but with different intensities and magnitudes. As shown in Table 4, between November 2004 and June 2005, the SSI12 computed using Gamma, Log-Normal, and Weibull PDFs produced severe ($-1.6$) drought conditions, while the SSI12 computed using Log-Logistic and PTIII PDFs produced moderate ($-1.4$) drought conditions. As shown in Table 5, between December 2015 and April 2016, the SSI12 computed using Gamma, Log-Normal, and Weibull PDFs produced extreme and severe ($-2.2$, $-2.3$, and $-2.0$, respectively) drought conditions, while the SSI12 computed using Log-Logistic and PTIII PDFs produced severe ($-1.6$) drought conditions.

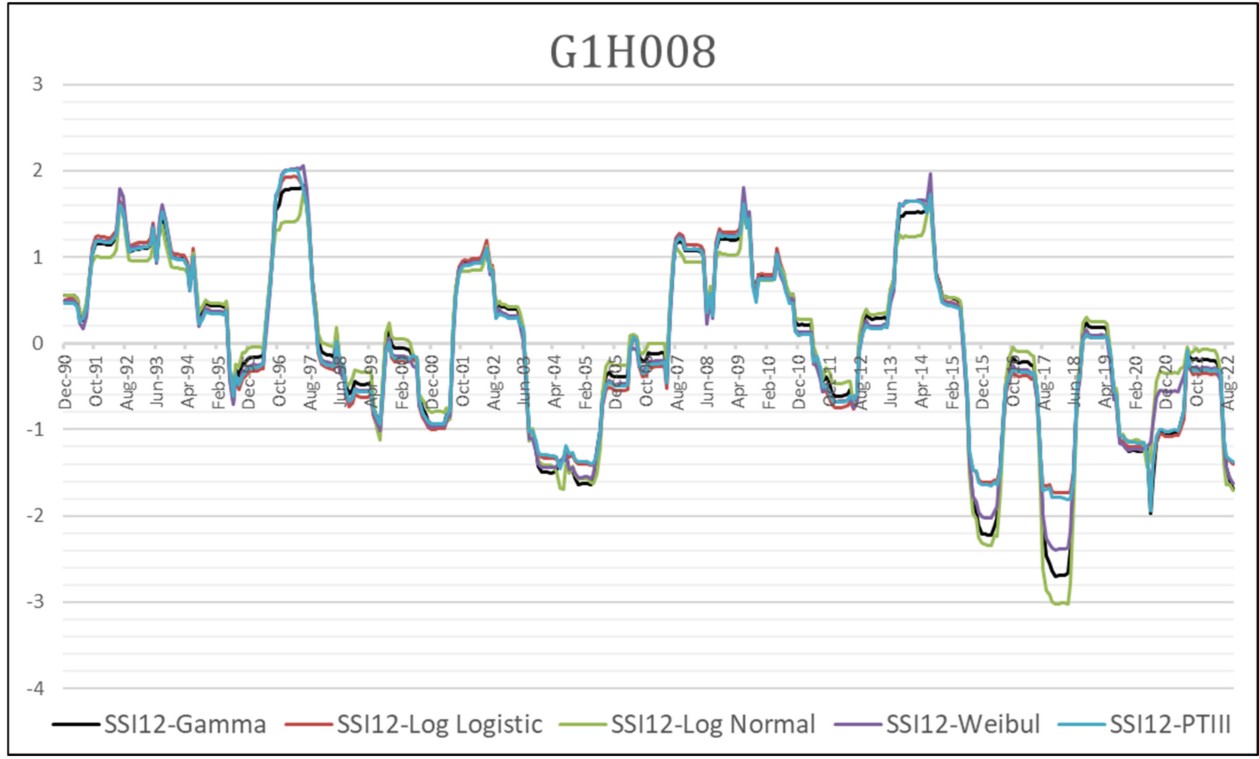

**Figure 4.** SSI12 results for G1H008 streamflow time series computed using Gamma, Log-Logistic, Log-Normal, PTIII, and Weibull Probability Distribution Functions.

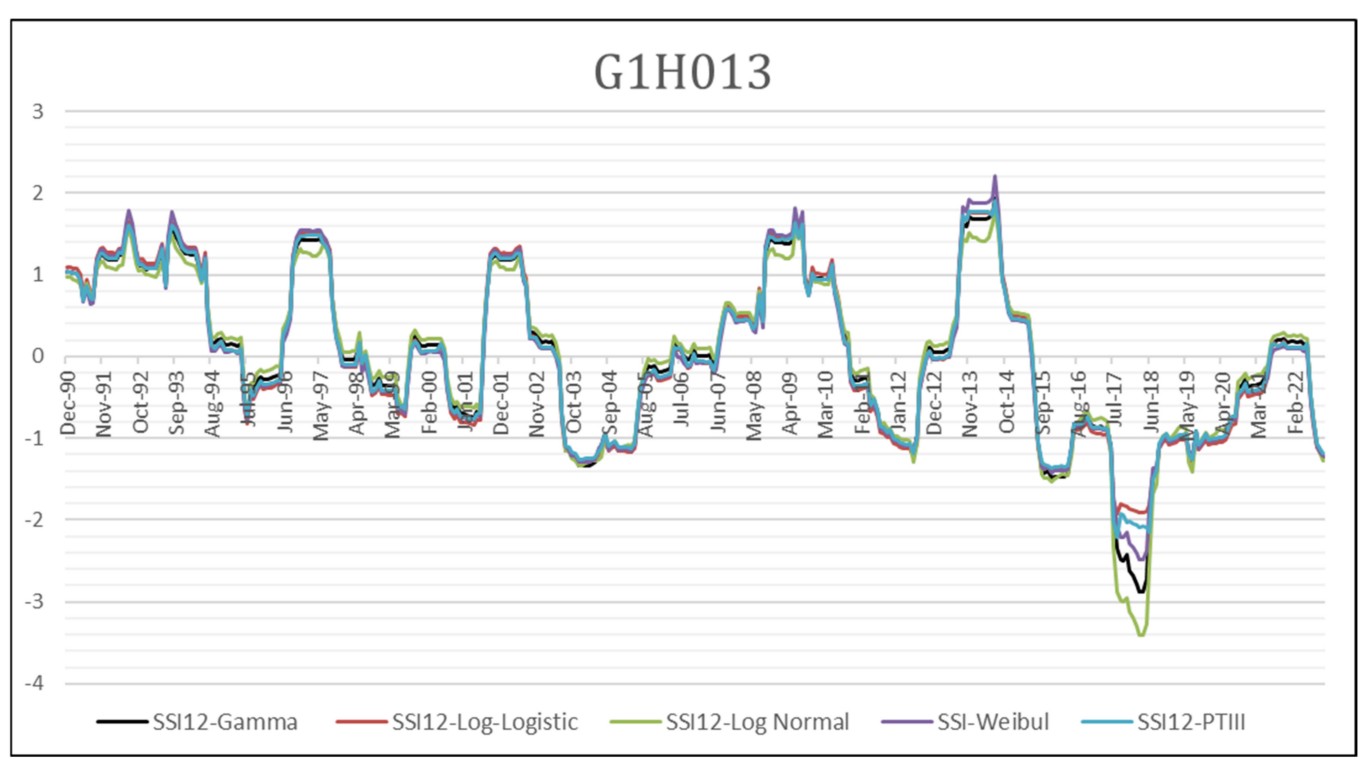

**Figure 5.** SSI12 results for G1H013 Streamflow time series computed using Gamma, Log-Logistic, Log-Normal, PTIII, and Weibull Probability Distribution Functions.

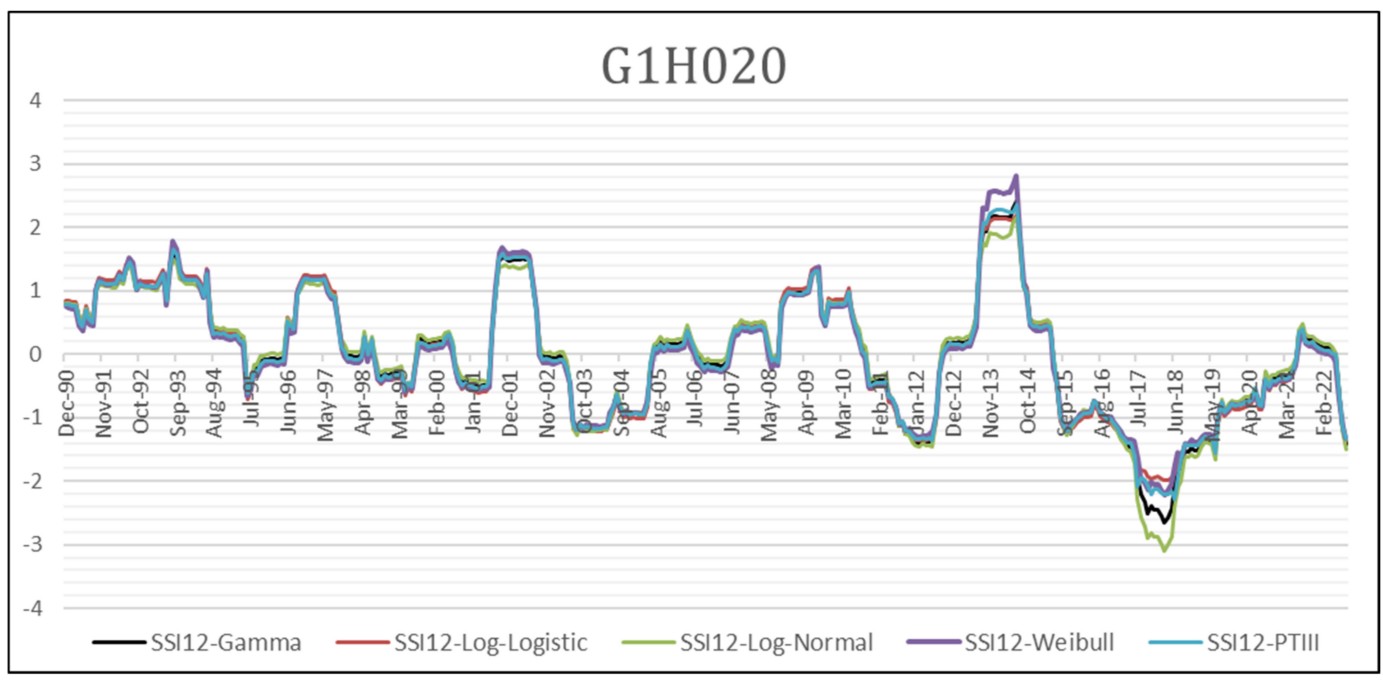

**Figure 6.** SSI12 results for G1H020 streamflow time series computed using Gamma, Log-Logistic, Log-Normal, PTIII, and Weibull Probability Distribution Functions.

**Table 4.** SSI12 computed using Gamma, Log-Logistic, Log-Normal, PTIII, and Weibull PDFs for the G1H008 station between November 2004 and June 2005.

| | G1H008 | | | | | | | | | |
|---|---|---|---|---|---|---|---|---|---|---|
| Month-Year | SSI12 Gamma | Drought Classification | SSI12 Log-Logistic | Drought Classification | SSI12 Log-Normal | Drought Classification | SSI12 PTIII | Drought Classification | SSI12 Weibull | Drought Classification |
| November 2004 | −1.6 | Severe | −1.4 | Moderate | −1.5 | Severe | −1.3 | Moderate | −1.5 | Severe |
| December 2004 to April 2005 | −1.6 | Severe | −1.4 | Moderate | −1.6 | Severe | −1.4 | Moderate | −1.6 | Severe |
| May 2005 | −1.6 | Severe | −1.4 | Moderate | −1.6 | Severe | −1.4 | Moderate | −1.5 | Severe |
| June 2005 | −1.6 | Severe | −1.4 | Moderate | −1.5 | Severe | −1.3 | Moderate | −1.5 | Severe |
| Average | −1.6 | Severe | −1.4 | Moderate | −1.6 | Severe | −1.4 | Moderate | −1.6 | Severe |

**Table 5.** SSI12 computed using Gamma, Log-Logistic, Log-Normal, PTIII, and Weibull PDFs for the G1H008 station between December 2015 and April 2016.

| | G1H008 | | | | | | | | | |
|---|---|---|---|---|---|---|---|---|---|---|
| Month-Year | SSI12 Gamma | Drought Classification | SSI12 Log-Logistic | Drought Classification | SSI1 Log-Normal | Drought Classification | SSI12 PTIII | Drought Classification | SSI12 Weibull | Drought Classification |
| December 2015 to March 2016 | −2.2 | Extreme | −1.6 | Severe | −2.3 | Extreme | −1.6 | Severe | −2.0 | Extreme |
| April 2016 | −2.1 | Extreme | −1.6 | Severe | −2.2 | Extreme | −1.6 | Severe | −1.9 | Severe |
| Average | −2.2 | Extreme | −1.6 | Severe | −2.3 | Extreme | −1.6 | Severe | −2.0 | Extreme |

Hence, from Figure 4 as well as Tables 4 and 5, it is apparent that the variability of drought intensity produced by the SSI12 computed using the Gamma, Log-Logistic, Log-Normal, PTIII and Weibull PDFs increases as drought conditions increase from moderate to extreme.

In Figure 5 (G1H013), it can be observed that the SSI12 time series computed using the Gamma, Log-Logistic, Log-Normal, PTIII, and Weibull PDFs produced drought events with similar onset and end times but with different intensities and magnitudes. This is a similar outcome to Figure 4 (G1H008). As shown in Table 6, between November 2003 and May 2005, the SSI12 calculated using Gamma, Log-Normal, Log-Logistic, PTIII, and Weibull PDFs all produced moderate (−1.2) drought conditions. As shown in Table 7, between August 2017 and May 2018, the SSI12 computed using Gamma, Log-Normal, PTIII, and Weibull PDFs produced extreme (−2.6, −3.2, −2.1, and −2.3, respectively) drought conditions, while the SSI12 computed using the Log-Logistic PDF produced severe (−1.9) drought conditions. Hence, from Figure 5 as well as Tables 6 and 7, it is apparent that the variability of drought intensity produced by the SSI12 computed using the Gamma, Log-Logistic, Log-Normal, PTIII, and Weibull PDFs increases as drought conditions increase from moderate to extreme. This is similar to the results obtained from station G1H008.

**Table 6.** SSI12 computed using Gamma, Log-Logistic, Log-Normal, PTIII, and Weibull PDFs for the G1H013 station between November 2003 and May 2005.

| | G1H013 | | | | | | | | | |
|---|---|---|---|---|---|---|---|---|---|---|
| Month-Year | SSI12 Gamma | Drought Classification | SSI12 Log-Logistic | Drought Classification | SSI12 Log-Normal | Drought Classification | SSI12 PTIII | Drought Classification | SSI12 Weibull | Drought Classification |
| November 2003 | −1.2 | Moderate | −1.2 | Moderate | −1.3 | Moderate | −1.2 | Moderate | −1.2 | Moderate |
| December 2003 to May 2004 | −1.3 | Moderate | −1.3 | Moderate | −1.3 | Moderate | −1.3 | Moderate | −1.3 | Moderate |
| June 2004 | −1.1 | Moderate | −1.2 | Moderate | −1.3 | Moderate | −1.2 | Moderate | −1.1 | Moderate |
| July 2004 | −1.1 | Moderate | −1.2 | Moderate | −1.2 | Moderate | −1.1 | Moderate | −1.1 | Moderate |
| August 2004 | −1.0 | Moderate | −1.0 | Moderate | −1.0 | Moderate | −1.0 | Moderate | −1.0 | Moderate |

**Table 6.** *Cont.*

| | G1H013 | | | | | | | | | |
|---|---|---|---|---|---|---|---|---|---|---|
| Month-Year | SSI12 Gamma | Drought Classification | SSI12 Log-Logistic | Drought Classification | SSI12 Log-Normal | Drought Classification | SSI12 PTIII | Drought Classification | SSI12 Weibull | Drought Classification |
| September 2004 | −1.1 | Moderate | −1.2 | Moderate | −1.1 | Moderate | −1.1 | Moderate | −1.1 | Moderate |
| October 2004 | −1.1 | Moderate | −1.1 | Moderate | −1.1 | Moderate | −1.1 | Moderate | −1.1 | Moderate |
| November 2004 | −1.0 | Moderate | −1.1 | Moderate | −1.0 | Moderate | −1.0 | Moderate | −1.0 | Moderate |
| December 2004 to April 2005 | −1.2 | Moderate | −1.2 | Moderate | −1.1 | Moderate | −1.1 | Moderate | −1.1 | Moderate |
| May 2005 | −1.2 | Moderate | −1.1 | Moderate | −1.0 | Moderate | −1.1 | Moderate | −1.1 | Moderate |
| Average | −1.2 | Moderate | −1.2 | Moderate | −1.2 | Moderate | −1.1 | Moderate | −1.2 | Moderate |

**Table 7.** SSI12 computed using Gamma, Log-Logistic, Log-Normal, PTIII, and Weibull PDFs for the G1H013 station between August 2017 and May 2018.

| | G1H013 | | | | | | | | | |
|---|---|---|---|---|---|---|---|---|---|---|
| Month-Year | SSI12 Gamma | Drought Classification | SSI12 Log-Logistic | Drought Classification | SSI12 Log-Normal | Drought Classification | SSI12 PTIII | Drought Classification | SSI12 Weibull | Drought Classification |
| August 2017 | −2.3 | Extreme | −1.9 | Severe | −2.9 | Extreme | −2.2 | Extreme | −2.1 | Extreme |
| September 2017 to October 2017 | −2.5 | Extreme | −1.8 | Severe | −3.0 | Extreme | −1.9 | Severe | −2.2 | Extreme |
| November 2017 | −2.4 | Extreme | −1.8 | Severe | −3.0 | Extreme | −2.0 | Extreme | −2.2 | Extreme |
| December 2017 | −2.6 | Extreme | −1.9 | Severe | −3.1 | Extreme | −2.0 | Extreme | −2.3 | Extreme |
| January 2018 | −2.7 | Extreme | −1.9 | Severe | −3.2 | Extreme | −2.0 | Extreme | −2.3 | Extreme |
| February 2018 | −2.8 | Extreme | −1.9 | Severe | −3.3 | Extreme | −2.1 | Extreme | −2.4 | Extreme |
| March 2018 to April 2018 | −2.9 | Extreme | −1.9 | Severe | −3.4 | Extreme | −2.1 | Extreme | −2.5 | Extreme |
| May 2018 | −2.7 | Extreme | −1.9 | Severe | −3.3 | Extreme | −2.1 | Extreme | −2.4 | Extreme |
| Average | −2.6 | Extreme | −1.9 | Severe | −3.2 | Extreme | −2.1 | Extreme | −2.3 | Extreme |

In Figure 6 (G1H020), it can be observed that the SSI12 time series computed using the Gamma, Log-Logistic, Log-Normal, PTIII, and Weibull PDFs produced drought events with similar onset and end times but with different intensities and magnitudes. This is a similar outcome to Figure 4 (G1H008) and Figure 5 (G1H013). As shown in Table 8, between July 2003 and June 2004, the SSI12 computed using Gamma, Log-Normal, Log-Logistic, PTIII, and Weibull PDFs all produced moderate (−1.2 for Gamma, Log-Normal, and Log-Logistic and −1.1 for PTIII and Weibull PDFs) drought conditions. As shown in Table 9, between July 2017 and May 2018, the SSI12 computed using Gamma, Log-Normal, PTIII and Weibull PDFs produced extreme (−2.4, −28, −2.1, and −2.0, respectively) drought conditions, while the SSI12 computed using the Log-Logistic PDF produced severe (−1.9) drought conditions. Hence, from Figure 6 as well as Tables 8 and 9, it is apparent that the variability of drought intensity produced by the SSI12 computed using the Gamma, Log-Logistic, Log-Normal, PTIII, and Weibull PDFs increases as drought conditions increase from moderate to extreme. This is similar to the results obtained from the G1H008 and G1H013 stations.

**Table 8.** SSI12 computed using Gamma, Log-Logistic, Log-Normal, PTIII, and Weibull PDFs for the G1H020 station between July 2003 and June 2004.

| | G1H013 | | | | | | | | | |
|---|---|---|---|---|---|---|---|---|---|---|
| Month-Year | SSI12 Gamma | Drought Classification | SSI12 Log-Logistic | Drought Classification | SSI12 Log-Normal | Drought Classification | SSI12 PTIII | Drought Classification | SSI12 Weibull | Drought Classification |
| July 2003 | −1.1 | Moderate | −1.2 | Moderate | −1.2 | Moderate | −1.1 | Moderate | −1.0 | Moderate |
| August 2003 | −1.2 | Moderate | −1.2 | Moderate | −1.3 | Moderate | −1.2 | Moderate | −1.2 | Moderate |

**Table 8.** *Cont.*

| | | | | | | | | | | |
|---|---|---|---|---|---|---|---|---|---|---|
| | | | | | **G1H013** | | | | | |
| **Month-Year** | **SSI12 Gamma** | **Drought Classifi-cation** | **SSI12 Log-Logistic** | **Drought Classifi-cation** | **SSI12 Log-Normal** | **Drought Classifi-cation** | **SSI12 PTIII** | **Drought Classifi-cation** | **SSI12 Weibull** | **Drought Classifi-cation** |
| September 2003 | −1.1 | Moderate | −1.1 | Moderate | −1.1 | Moderate | −1.1 | Moderate | −1.1 | Moderate |
| October 2003 | −1.2 | Moderate | −1.2 | Moderate | −1.2 | Moderate | −1.2 | Moderate | −1.2 | Moderate |
| November 2003 | −1.2 | Moderate | −1.2 | Moderate | −1.2 | Moderate | −1.1 | Moderate | −1.1 | Moderate |
| December 2003 to May 2004 | −1.2 | Moderate | −1.2 | Moderate | −1.2 | Moderate | −1.2 | Moderate | −1.1 | Moderate |
| June 2004 | −1.0 | Moderate | −1.0 | Moderate | −1.0 | Moderate | −1.0 | Moderate | −0.9 | Mild |
| Average | −1.2 | Moderate | −1.2 | Moderate | −1.2 | Moderate | −1.1 | Moderate | −1.1 | Moderate |

**Table 9.** SSI12 computed using Gamma, Log-Logistic, Log-Normal, PTIII, and Weibull PDFs for the G1H020 station between July 2017 and May 2018.

| | | | | | | | | | | |
|---|---|---|---|---|---|---|---|---|---|---|
| | | | | | **G1H013** | | | | | |
| **Month-Year** | **SSI12 Gamma** | **Drought Classifi-cation** | **SSI12 Log-Logistic** | **Drought Classifi-cation** | **SSI12 Log-Normal** | **Drought Classifi-cation** | **SSI12 PTIII** | **Drought Classifi-cation** | **SSI12 Weibull** | **Drought Classifi-cation** |
| July 2017 | −1.9 | Severe | −1.8 | Severe | −2.3 | Extreme | −2.1 | Extreme | −1.7 | Severe |
| August 2017 | −2.2 | Extreme | −1.8 | Severe | −2.6 | Extreme | −2.0 | Extreme | −1.9 | Severe |
| September 2017 | −2.3 | Extreme | −1.9 | Severe | −2.7 | Extreme | −2.0 | Extreme | −2.0 | Extreme |
| October 2017 | −2.5 | Extreme | −1.9 | Severe | −2.9 | Extreme | −2.1 | Extreme | −2.1 | Extreme |
| November 2017 | −2.4 | Extreme | −2.0 | Severe | −2.8 | Extreme | −2.2 | Extreme | −2.0 | Extreme |
| December 2017 to January 2018 | −2.4 | Extreme | −1.9 | Severe | −2.9 | Extreme | −2.1 | Extreme | −2.1 | Extreme |
| February 2018 | −2.6 | Extreme | −2.0 | Extreme | −3.0 | Extreme | −2.2 | Extreme | −2.1 | Extreme |
| March 2018 | −2.7 | Extreme | −2.0 | Extreme | −3.1 | Extreme | −2.2 | Extreme | −2.2 | Extreme |
| April 2018 | −2.6 | Extreme | −2.0 | Extreme | −3.0 | Extreme | −2.2 | Extreme | −2.2 | Extreme |
| May 2018 | −2.4 | Extreme | −1.9 | Severe | −2.9 | Extreme | −2.2 | Extreme | −2.1 | Extreme |
| Average | −2.4 | Extreme | −1.9 | Severe | −2.8 | Extreme | −2.1 | Extreme | −2.0 | Extreme |

*4.2. The S-W Test for Normality on the SSI Calculated Using the Selected PDFs*

Thus far, the SSI12 results for the three selected streamflow gauging stations (G1H008, G1H013, and G1H020) have shown that the Gamma, Log-Normal, Log-Logistic, PTIII, and Weibull PDFs produced comparable results for mild (0.00 to −0.99) to moderate (−1.00 to −1.49) drought conditions. For severe (−1.5 to −1.99) to extreme (≤−2.0), there remains uncertainty on the choice of a suitable PDF for SSI calculation due to the increased variability in drought intensity produced by the different PDFs. To determine which PDF is most suitable for SSI computation for severe to extreme drought conditions, the S-W test for normality was used. The aim was to use the S-W test for normality to assess and select the best-fitting PDF between the Gamma, Log-Normal Log-Logistic, PTIII, and Weibull PDFs. As shown in Table 10, none of the PDFs met the S-W condition for normality. Hence, the S-W normality test results were inconclusive for the considered PDFs.

*4.3. Visual Inspection of the SSI Calculated Using the Selected PDFs*

In the absence of conclusive S-W results to aid in the selection of the most suitable PDFs for SSI calculation in the BRC, a visual inspection of the SSI calculated using the selected PDFs was carried out to identify any obvious systematic differences or similarities. This visual inspection approach is not uncommon in this type of research; it was employed by Svensson et al. (2017) [20]. As shown in the example in Figure 7, it was observed visually that the Gamma, Log-Normal, and Weibull PDFs were the only PDFs that were able to identify extreme (≤2.0) drought conditions for the G1H008 between July 2015 and July 2018. Visual inspection of Figures 8–10 shows that the SSI calculated using Gamma,

Log-Normal, and Weibull PDFs were the only PDFs that were able to consistently identify extreme ($\leq 2.0$) drought conditions for the G1H008, G1H013, and G1H020. On the other hand, the SSI calculated using PTIII and Log-Logistic failed to consistently identify extreme drought conditions for G1H008, G1H013, and G1H020.

**Table 10.** Shapiro-Wilk Normality Test results for SSI$-12$ time series calculated using Gamma, Log-Logistic, Log-Normal, PTIII, and Weibull PDFs on streamflow gauging stations G1H008, G1H013 and G1H020.

| | Shapiro-Wilk Test for Normality | | | | |
|---|---|---|---|---|---|
| | **Gamma** | **Log-Logistic** | **PTIII** | **Log-Normal** | **Weibull** |
| G1H008 | W = 0.97464 $p$-value = $3.06\times 10^{-6}$ | W = 0.96785 $p$-value = $1.83\times 10^{-7}$ | W = 0.97958 $p$-value = $3.062\times 10^{-5}$ | W = 0.94277 $p$-value = $5.416\times 10^{-11}$ | W = 0.94277 $p$-value = $5.416\times 10^{-11}$ |
| G1H013 | W = 0.9717 $p$-value = $8.386\times 10^{-7}$ | W = 0.95804 $p$-value = $5.186\times 10^{-9}$ | W = 0.97103 $p$-value = $6.346\times 10^{-7}$ | W = 0.94802 $p$-value = $2.293\times 10^{-10}$ | W = 0.97389 $p$-value = $2.14\times 10^{-6}$ |
| G1H020 | W = 0.99019 $p$-value = 0.01188 | W = 0.98243 $p$-value = 0.0001341 | W = 0.99018 $p$-value = 0.01177 | W = 0.97238 $p$-value = $1.186\times 10^{-6}$ | W = 0.98534 $p$-value = 0.0006527 |

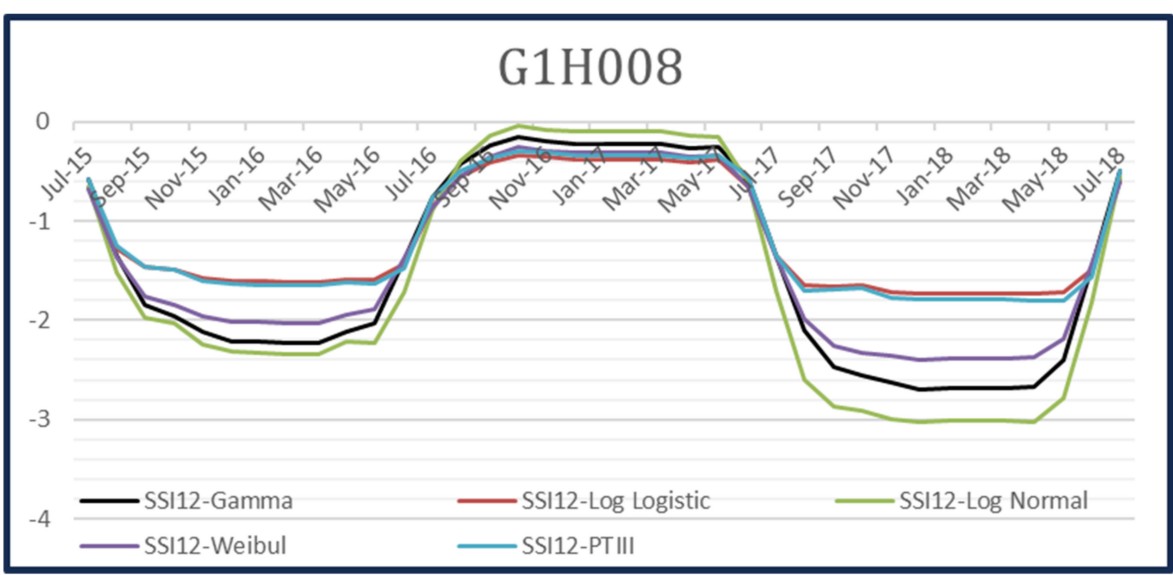

**Figure 7.** Example of visual inspection of the SSI12 calculated using the Gamma, Log-Logistic, Log-Normal, PTIII, and Weibull for the G1H008 streamflow gauging station; from July 2015 to July 2018.

*4.4. Evaluation of the Correlation between the SSI Computed Using the Selected PDFs*

A correlation statistical test was performed on the SSI time series computed using the Gamma, Log-Normal, PTIII, Log-Logistic, and Weibull PDFs to assess the extent of the similarities or differences amongst them. As shown in Table 11, the correlation statistics show that they produced a positive correlation, mostly above 99%. Hence, they produced relatively similar SSI time series. The positive correlation is an indication they all produced similar major and minor drought conditions. In Table 11, the correlation statistics for the SSI12 time series for G1H008, G1H013, and G1H020 computed using the Gamma PDF and SPI12 for Franschoek produced a positive correlation, mostly between 75% and 83%. The positive correlation is an indication that they all produced similar major drought conditions.

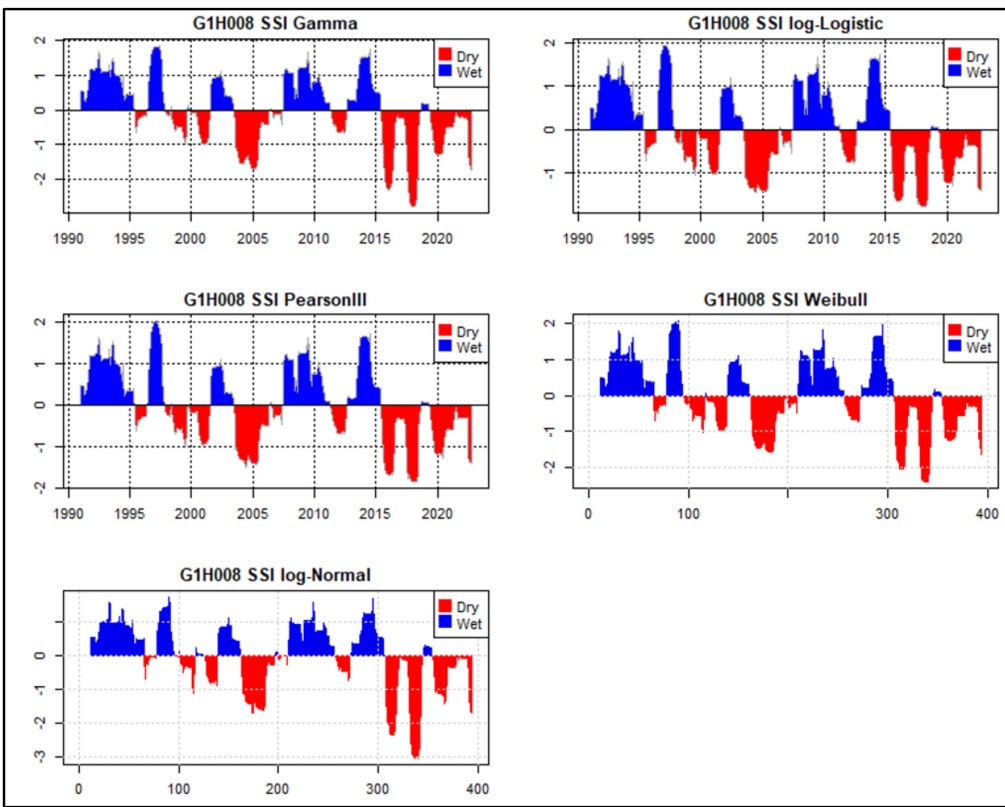

**Figure 8.** SSI12 time series for G1H008 computed using Gamma, PTIII (PearsonIII), Log-Normal, Log-Logistic, and Weibull PDFs in the Berg River Catchment.

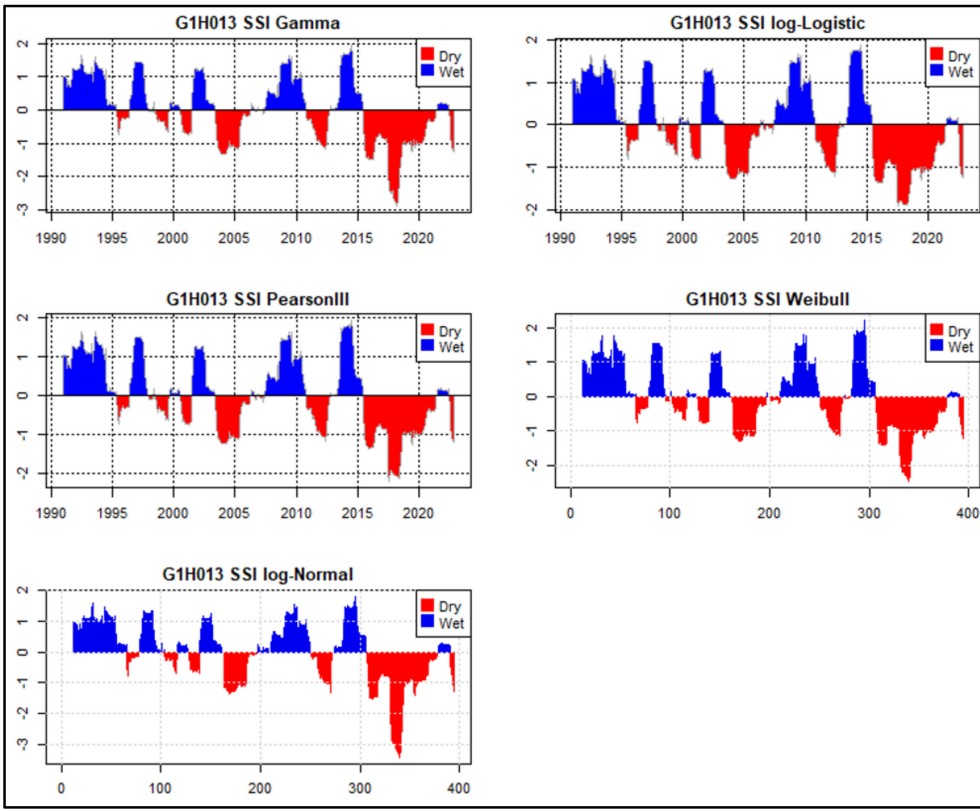

**Figure 9.** SSI12 time series for G1H013 calculated using Gamma, PTIII (PearsonIII), Log-Normal, Log-Logistic, and Weibull PDFs in the Berg River Catchment.

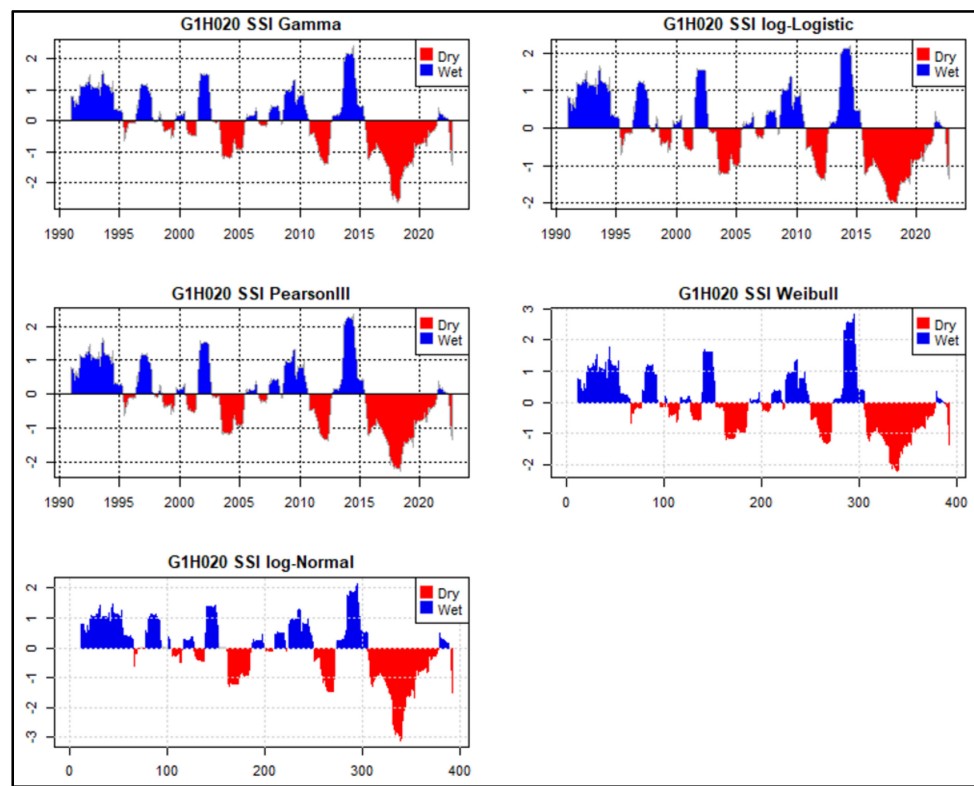

**Figure 10.** SSI12 time series for G1H020 computed using Gamma, PTIII (PearsonIII), Log-Normal, Log-Logistic, and Weibull PDFs in the Berg River Catchment.

**Table 11.** Correlation statistics for the SSI time series computed using the Gamma, Log-Normal, PTIII, Log-Logistic, and Weibull.

| G1H008 | | | | | | |
|---|---|---|---|---|---|---|
| | **SSI Gamma** | **SSI log-Logistic** | **SSI log-Normal** | **SSI PTIII** | **SSI Weibull** | **Franschoek SPI12 (Gamma)** |
| SSI Gamma | 1 | | | | | |
| SSI log-Logistic | 0.98223 | 1 | | | | |
| SSI log-Normal | 0.984512 | 0.947839 | 1 | | | |
| SSI PTIII | 0.986172 | 0.998936 | 0.953691 | 1 | | |
| SSI Weibull | 0.991315 | 0.98774 | 0.977377 | 0.990254 | 1 | |
| Franschoek SPI12 (Gamma) | 0.741493 | 0.750306 | 0.715999 | 0.752611 | 0.754136 | 1 |
| G1H013 | | | | | | |
| | SSI Gamma | SSI log-Logistic | SSI log-Normal | SSI PTIII | SSI Weibull | Franschoek SPI12 (Gamma) |
| SSI Gamma | 1 | | | | | |
| SSI log-Logistic | 0.989213 | 1 | | | | |
| SSI log-Normal | 0.991976 | 0.965913 | 1 | | | |
| SSI PTIII | 0.994384 | 0.998255 | 0.976084 | 1 | | |
| SSI Weibull | 0.995609 | 0.995802 | 0.977003 | 0.998044 | 1 | |
| Franschoek SPI12 (Gamma) | 0.803445 | 0.81042 | 0.782366 | 0.811022 | 0.810208 | 1 |

**Table 11.** *Cont.*

| | G1H008 | | | | | |
|---|---|---|---|---|---|---|
| | SSI Gamma | SSI log-Logistic | SSI log-Normal | SSI PTIII | SSI Weibull | Franschoek SPI12 (Gamma) |
| | G1H020 | | | | | |
| | SSI Gamma | SSI log-Logistic | SSI log-Normal | SSI PTIII | SSI Weibull | Franschoek SPI12 (Gamma) |
| SSI Gamma | 1 | | | | | |
| SSI log-Logistic | 0.994149 | 1 | | | | |
| SSI log-Normal | 0.994499 | 0.980957 | 1 | | | |
| SSI PTIII | 0.997659 | 0.997659 | 0.987417 | 1 | | |
| SSI Weibull | 0.992818 | 0.993924 | 0.975533 | 0.996049 | 1 | |
| Franschoek SPI12 (Gamma) | 0.817348 | 0.827468 | 0.804458 | 0.82031 | 0.816487 | 1 |

*4.5. Comparison of the SSI with SPI Results*

The credibility of the SSI12 time series obtained using both the Gamma, Lo-Logistic, PTIII, Log-Normal, and Weibull PDFs was tested by comparing it with an SPI12 time series obtained using the Gamma PDF for both the Franschoek and Stellenbosch stations. As shown in Figure 11, the Comparison between the SPI12 time series for Franschoek and the SPI12 time series for Stellenbosch reveals that both SPI time series are closely similar. Hence, since Franschoek is located within the BRC, the Franschoek SPI 12 time series was used to test the credibility of the SSI12 for G1H008, G1H013, and G1H020. The SSI12 time series for G1H008, G1H013, and G1H020 computed using the Gamma PDF were used for comparisons with the SPI 12 for Franschoek. For instance, both the SSI12 and SPI 12 identify the extreme drought conditions that occurred during the 2015–2018 period.

In Figures 12–14, it can be observed that both the SSI12 for G1H008, G1H013, and G1H020 computed using the Gamma PDF are closely similar to the SPI12 for Franschoek in that they both produced or identified all the major drought events that occurred between the years 1990 and 2022.

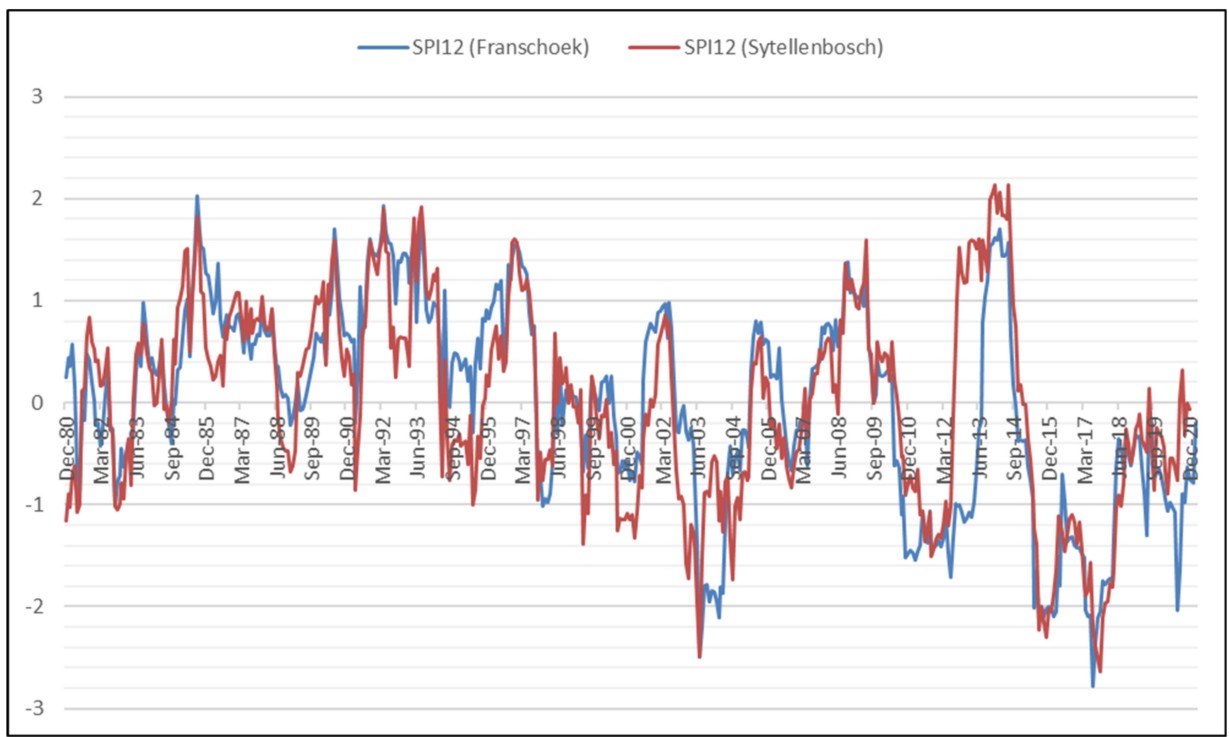

**Figure 11.** Comparison between SPI12 time series for Franschoek and SPI12 time series for Stellenbosch.

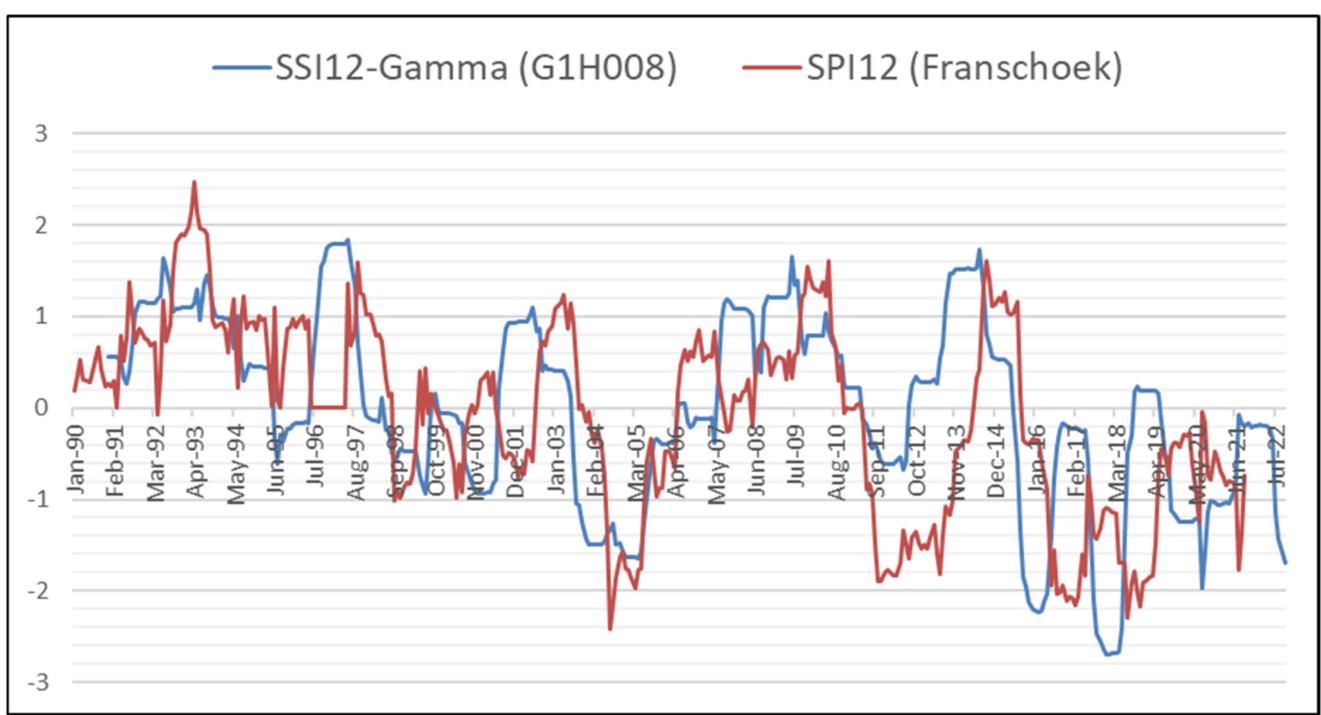

**Figure 12.** Comparison between SPI12 time series for Franschoek and SSI12 time series for G1H008.

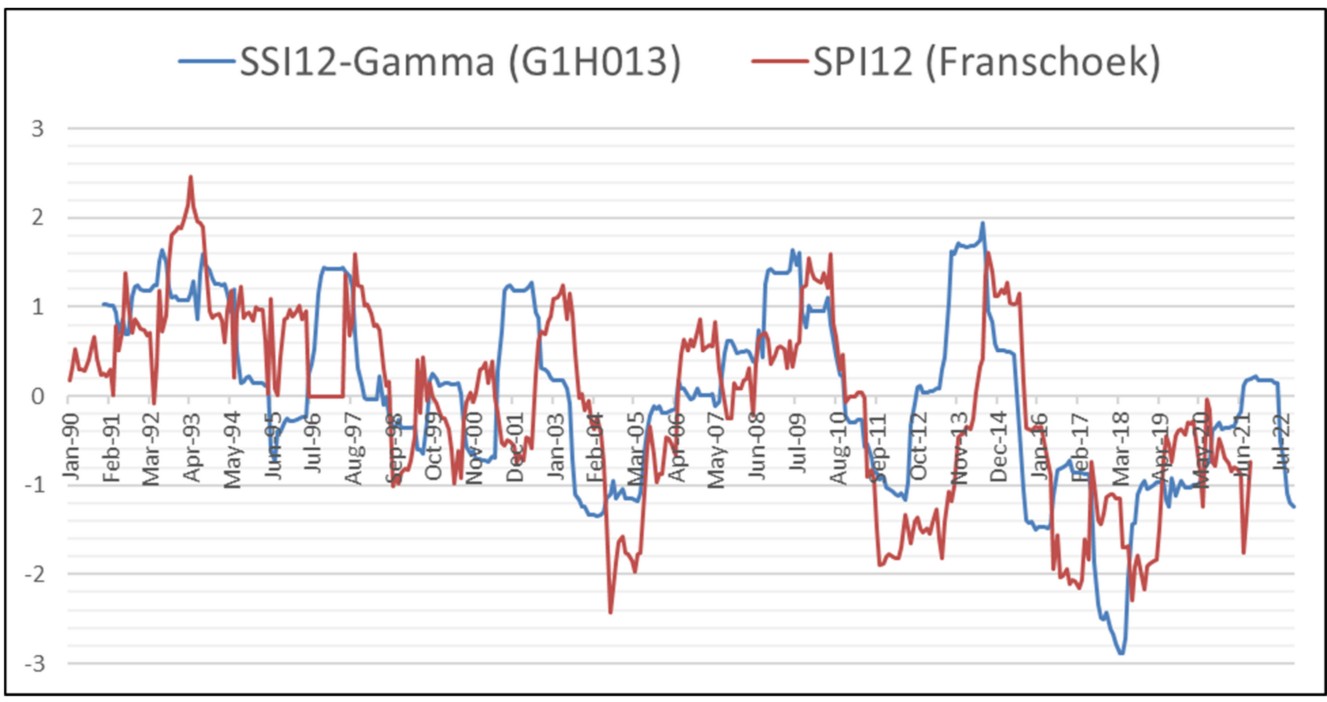

**Figure 13.** Comparison between SPI12 time series for Franschoek and SSI12 time series for G1H013.

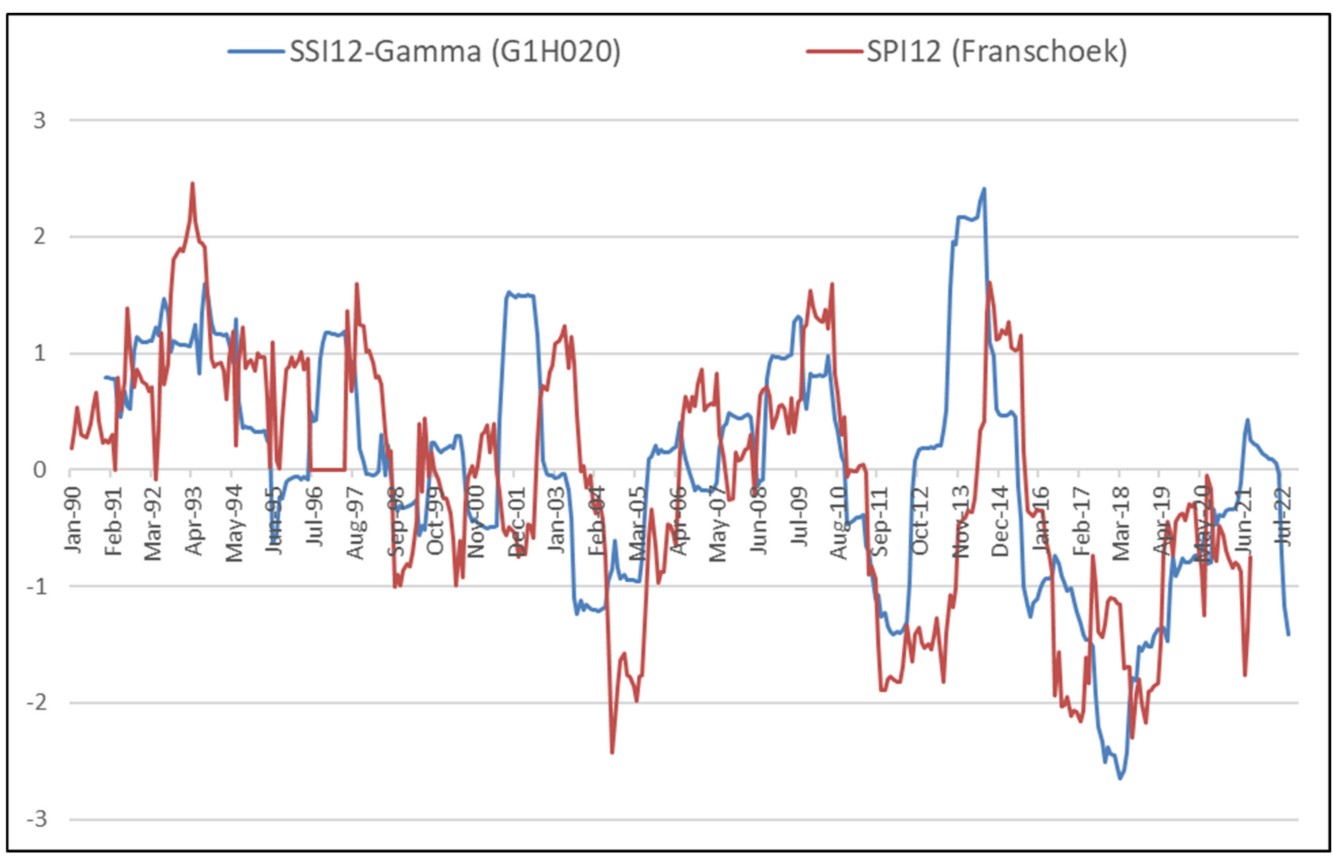

**Figure 14.** Comparison between SPI12 time series for Franschoek and SSI12 time series for G1H020.

*4.6. Drought Assessment Using the SSI Calculated Using the Gamma, Log-Normal and Weibull PDFs*

As shown in Figures 8–10 and Tables 12–14, historical drought assessment using the SSI12 time series calculated using the Gamma, Log-Normal, and Weibull PDFs in the BRC indicates that for both G1H008, G1H013, and G1H020, drought events have been occurring with more intensity between the years 1990 and 2022; the most severe were during the 2015–2018 extreme drought condition.

**Table 12.** Drought assessment during the period between 1990 and 2022 using the SSI12 calculated using the recommended Gamma, Log-Normal, and Weibull PDFs for the G1H008 streamflow gauging station.

| Streamflow Gauging Station | Drought Period | Average SSI12 | | | Drought Classification |
|---|---|---|---|---|---|
| | | Gamma | Log-Normal | Weibull | |
| G1H008 | June 2000 to June 2001 | −0.7 | −0.7 | −0.8 | Mild Drought |
| | September 2004 to May 2005 | −1.6 | −1.6 | −1.5 | Severe Drought |
| | September 2015 to April 2016 | −2.1 | −2.2 | −2.0 | Extreme Drought |
| | August 2017 to May 2018 | −2.6 | −2.9 | −2.3 | Extreme Drought |
| | September 2019 to July 2020 | −1.3 | −1.3 | −1.2 | Moderate Drought |

**Table 13.** Drought assessment during the period between 1990 and 2022 using the SSI12 calculated using the recommended Gamma, Log-Normal, and Weibull PDFs for the G1H013 streamflow gauging station.

| Streamflow Gauging Station | Drought Period | Average SSI12 | | | Drought Classification |
| --- | --- | --- | --- | --- | --- |
| | | Gamma | Log-Normal | Weibull | |
| G1H013 | August 2000 to June 2001 | −0.6 | −0.6 | −0.7 | Mild Drought |
| | August 2003 to May 2005 | −1.2 | −1.2 | −1.2 | Moderate Drought |
| | October 2011 to July 2012 | −1.0 | −1.0 | −1.0 | Moderate Drought |
| | August 2015 to June 2016 | −1.4 | −1.4 | −1.3 | Moderate Drought |
| | July 2017 to June 2018 | −2.5 | −3.0 | −2.2 | Extreme Drought |
| | July 2018 to February 2019 | −1.1 | −1.1 | −1.1 | Moderate Drought |

**Table 14.** Drought assessment during the period between 1990 and 2022 using the SSI12 calculated using the recommended Gamma, Log-Normal, and Weibull PDFs for the G1H020 streamflow gauging station.

| Streamflow Gauging Station | Drought Period | Average SSI12 | | | Drought Classification |
| --- | --- | --- | --- | --- | --- |
| | | Gamma | Log-Normal | Weibull | |
| G1H020 | August 2000 to June 2001 | −0.4 | −0.4 | −0.5 | Mild Drought |
| | July 2003 to June 2004 | −1.2 | −1.2 | −1.1 | Moderate Drought |
| | August 2011 to July 2012 | −1.3 | −1.3 | −1.2 | Moderate Drought |
| | August 2015 to January 2016 | −1.1 | −1.1 | −1.1 | Moderate Drought |
| | July 2017 to June 2018 | −2.4 | −2.8 | −2.0 | Extreme Drought |

## 5. Discussion

Despite the urgent need to improve drought monitoring using streamflow-based indices, insufficient studies have been conducted to test the applicability of the SSI for hydrological drought monitoring in SA. The few studies that exist have not tested the sensitivity of the SSI to various commonly used PDFs to recommend the most suitable PDFs [38]. Thus, this study has investigated the applicability of the SSI as well as its sensitivity to the Gamma, Log-Logistic, Log-Normal, PTIII, and Weibull PDFs in the BRC, located in the WC province of SA. Streamflow time series spanning more than 30 years were acquired and used to compute the SSI, accumulated over a 12-month period (SSI12). The study has found that all the SSI12 computed using all the selected PDFs consistently produced drought events with similar onset and end times but with different intensities and magnitudes. The variability in drought intensity was more evident in the severe to extreme drought conditions. While the SSI calculated using the Gamma, Log-Logistic, Log-Normal, PTIII, and Weibull PDFs were able to detect all the mild to severe droughts during the study period, only the SSI computed using the Gamma, Log-Normal, and Weibull PDFs could detect the extreme drought conditions in the BRC. This is evidence that not all PDFs are suitable for SSI calculation in the BRC, and possibly in many other catchments in SA. Studies such as Botai et al. (2021) used the Gamma PDF to compute the SSI for drought monitoring in the WC province, on the basis that the Gamma is most commonly used [38]. The SSI results from this study have demonstrated that it is essential that various PDFs be tested before being accepted and used for SSI computation and application in SA catchments. The Gamma PDF cannot be universally accepted for SSI calculation as it is for SPI calculation.

To propose the most suitable PDF for SSI calculation and application in the BRC, the S-W test for normality was employed. However, the S-W test produced inconclusive

results, so the visual inspection approach was resorted to. Through visual inspection of the SSI computed using all the selected PDFs, it was observed that only the SSI computed using the Gamma, Log-Normal, and Weibull could detect the only extreme drought conditions (2015–2018) that occurred during the 1990–2022 study period. Therefore, this study discourages the use of the Log-Logistic and PTII PDFs to calculate the SSI for the BRC catchment due to their failure to detect the 2015–2018 extreme drought conditions. Furthermore, using the SSI computed using the Gamma, Log-Normal, and Weibull, this study has found that droughts have been occurring with increased intensity and that the detected 2015–208 extreme drought events are the worst streamflow-based hydrological droughts in the BRC during the 1990–2022 study period. This agrees with Botai et al. (2021), who, from their study in the WC province, concluded that the duration and severity of drought conditions over the WC province have increased during the 1985–2020 period, and identified the 2015–2020 drought as the worst during the 1985–2020 period. Botai attributed the increasing drought conditions in the WC to reduced streamflow, influenced by reduced precipitation or a shift in seasonal precipitation, coupled with increased temperature [38]. The study results also agree with other drought reports in the WC. For instance, Brühl and Visser (2021) reported the 2016–2018 WC drought as the worst drought in 100 years. It led to the anticipation of the so-called 'day zero', described as a situation in which the city of Cape Town would be left with only 10% of available water for human consumption [49]. Hence, this study recommends the Gamma, Log-Normal, and Weibull PDFs for computation and application of the SSI in the BRC catchment. Since the selected streamflow gauging stations are well spatially distributed across the BRC, the SSI based on the Gamma, Log-Normal, and Weibull PDFs is recommended for application throughout the BRC catchment.

The results from the evaluation of the correlation amongst the SSI computed using the selected PDFs and with the SPI showed that there are good similarities amongst the SSIs as well as between the SSI and the SPI. Comparison of the SSI with the SPI has also shown that both the SSI and SPI identify all the major droughts, including the 2015–2018 extreme drought event. Hence, this study has shown that the SSI is capable of characterizing streamflow-based hydrological drought in the BRC. The close correlation between the SSI12 and the SPI12 is an indication that the streamflow-based hydrological drought may be caused by climate factors such as precipitation deficit, in concurrence with Botai et al. (2021) [38]. Studies have shown the importance of using the SSI and the SPI to obtain the Propagation Threshold (PT) from meteorological drought to hydrological drought. This helps to provide early warning information for hydrological drought, which is vital for drought preparedness and mitigation [4]. This study thus proposes that future research should focus on more investigations using the SSI and the SPI to study the propagation of drought from meteorological to hydrological in the BRC and other catchments in SA.

## 6. Conclusions

To contribute to the provision of tested scientific knowledge on the effective application of the SSI in SA, this study has investigated the application of the SSI for hydrological drought monitoring in the BRC and WC provinces of SA. Using more than 30-year records of streamflow data (G1H008, G1H013, and G1H020) from the BRC, as well as five PDFs (PTIII, Log-Normal, Log-Logistic, Weibull, and Gamma), 12 months' SSI (SSI12) time series were computed and analyzed. The study has found that all the SSI time series computed using all the selected PDFs detected mild to extreme drought conditions with varying intensities and magnitudes during the 1990–2022 study period. It is therefore recommended that different PDFs, including those not tested in this study, always be tested before they are accepted and used for SSI computation and application in all SA catchments. On the basis that only the SSI time series computed using the Gamma, Log-Normal, and Weibull PDFs detected the 2015–2018 extreme droughts events, the Gamma, Log-Normal, and Weibull PDFs are recommended for SSI computation and application in the BRC. The 2015–2018 extreme drought in the BRC has been reported by other studies to have been the worst drought in almost 100 years. Comparison of the SSI12 (Gamma) with the SPI12

(Gamma) has provided evidence that the SSI is credible and is applicable for hydrological drought monitoring in the BRC. Both the SSI and the SPI identified all the major drought events during the study period, including the 2015–2018 extreme drought event. Based on these outcomes, it is recommended that further studies be conducted to investigate the propagation and evolution of drought from meteorological to hydrological drought using the SSI and SPI. These investigations will result in improved early warning information for hydrological drought in the BRC and other catchments in SA.

When comparing the study outcomes with those of other studies, it can be concluded that the detected droughts during the 1990–2022 study period are caused largely by climate-related factors such as precipitation deficit and increased evaporation. Thus, in anticipation of more frequent and intense droughts due to climate change factors, it is recommended that water resource managers take proactive action in searching for strategies to improve water resource management and drought preparedness, mitigation, and response in the WC province of SA. The application of the SSI for hydrological drought monitoring is relatively new in SA. Hence, this study has provided a foundation for more research on the application of the SSI in the WC and other catchments in SA.

**Author Contributions:** Conceptualization and methodology: M.B.M.; Software: N.S.M.; validation, formal analysis, and data curation: M.B.M., T.K., D.K. and N.S.M.; Original draft preparation: M.B.M.; Review and editing: M.B.M.; Supervision: T.K. and D.K. All authors have read and agreed to the published version of the manuscript.

**Funding:** This research was funded by the Department of Water and Sanitation, South Africa.

**Informed Consent Statement:** Informed consent was obtained from all subjects involved in the study.

**Data Availability Statement:** Not applicable.

**Acknowledgments:** The Department of Water and Sanitation, South Africa is acknowledged for providing funding to publish the article.

**Conflicts of Interest:** The authors declare no conflict of interest.

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
