# Peer review of "Application of the Standardised Streamflow Index for Hydrological Drought Monitoring in the Western Cape Province, South Africa: A Case Study in the Berg River Catchment"

_water, doi:10.3390/w15142530_

Round 1
Reviewer 1 Report
The paper is very important. However, authors need to improve the folowing:
1. The literature review/background is restricted to RSA. More annotated comments in pdf.
2. Having chosen the the best pdf for each series and station, i expected a detailed analysis of the best pdf per series
3. Justify why L-moments and the pdfs were pre-selected
4. Table 3 is not important. just cite the refs of the packages. Codes are found in the manuals of each package.
5. Figures 4, 5 and 6 can be improved for better visuality.
6. The Correlation Coefficient was not introduced in the methods.
7. Draw conclusions from the study. It is too general.

Language is fine.
Reviewer 2 Report
The manuscript contains a scientifically sound and statistically adequate analysis of hydrological drought events using the Standardized Streamflow Index in a region that has recently been experiencing recurrent drought episodes. It provides a useful, first-order assessment of the applicability of SSI and tests the sensitivity of various PDFs to affirm their performances. The introduction and the methods are well described with key findings herein that will appeal to climate change-interested audiences for this region. The work is relevant not just for the journal but to the area of study given the region is prone to extreme climate events and vulnerable to their impacts. Generally, the article is well written. However, the work requires some changes/clarifications to improve its quality/readership. My comments are as follows:
1. The abstracts need to be improved to have the flow describing brief background (well captured), the methods which describe the research design and data used, and now key findings which highlight the key outcomes of the study. The last paragraph states the main conclusion and implication of the study. As it is in its current form, the methods and results are intertwined thereby making it a challenge to have the flow of the very important subsection of this article.
2. Equations 2 - 6 could be summarized in a tabular manner (see random example; DOI 10. 1007/s12205-023-1423-z).
3. Table 3 should be transferred to supplementary material and not as part of the main manuscript.
4. The results should be improved by stating quantitative values in addition to the qualitative findings stated. See the example in lines 358-359.
5. I suggest separating the conclusion from the discussion and having the discussion section strengthen to shade more light beyond the content captured and reference stated in the current version of the manuscript. I would strongly recommend the authors to improve the discussion section. While improving the discussion section, briefly reiterate for readers the research problem or problems you are investigating and the methods used to investigate them. Then move quickly to describe the major findings of the study. You should write direct, declarative, and succinct proclamations of the study. Then go ahead and explain the meaning of the findings and why they are important. No one has thought as long and hard about your study as you have. Systematically explain the meaning of the findings and why you believe they are important. These findings should be discussed in the context of other existing similar studies. Lastly. Acknowledge the limitation of the studies and make suggestions for future studies.
Authors are advised to cross-check minor grammar issues in the text.
Round 2
Reviewer 1 Report
No more comments. Paper can be accepted
Author Response
Noted.
Reviewer 2 Report
I thank the authors for providing a prompt response. I appreciate the time spent by the authors to answer my remarks and improve their paper. Clearly, the abstract is now well drafted, and substantial effort made to see that the results section is improved. However, the manuscript is lacking in some aspects, which ends up weakening it too much. Several aspects must be improved:
Firstly, upon re-reading the introduction section, I noticed some inconsistencies in this section. I suggest the authors have a look at it and improve accordingly. The flow of this section should start with information about the problem to be solved (research gap) which you stated well. The next paragraph should capture the existing literature, then justifications. After this, the subsequent paragraph captures the main limitations (scope of the study), and then what you hope to achieve (core objectives). Lastly, is the structure of the paper. Now, reading the work, the paragraph in lines 143 to 180 ought to be captured before line 136. The last paragraph should clearly highlight the two objectives that the study will contribute to solving (i.e., to evaluate the applicability of SSI and to test the sensitivity of the SSI to different PDFs), and then have the benefits as indicated in line 197 all the way to the last line.
Secondly, I requested that the authors have a new sub-section that clearly captures the discussions to support the relevant findings of this study and even provided the basic structure of information that would guide the authors to draft this subsequent. On reading the revised draft, I couldn't locate such sections even as they claimed that I read sections 4 and 5. Please, in your response to the reviewer, highlight in your rebuttal exact lines and even copy the new changes in the response document.
Not an expert in this section. The authors should ensure that there are no grammatical syntax and typographies in the final text.
Author Response
Please see the attachment.
Replaced figures 4 to 11 with figures with improved resolution and quality.
